# A Closer Look at Effective Intervention Methods to Reduce Household Solid Waste Generation in Japan

Yoshinori Saitoh [1,2,*], Hiroshi Tago [2], Kimiyo Kumagai [2] and Akihiro Iijima [1]

1   Graduate School of Regional Policy, Takasaki City University of Economics, 1300, Kaminamie, Takasaki 370-0801, Japan
2   Gunma Prefectural Institute of Public Health and Environment Sciences, 378, Kamioki, Maebashi 371-0052, Japan
*   Correspondence: td19002sy@tcue.ac.jp or sai-yosi@pref.gunma.lg.jp

**Abstract:** In many countries municipal solid waste (MSW) is expected to soon increase beyond the pace of population growth due to urbanization. To minimize its negative impact, MSW management needs to be advanced. We studied administrative awareness-raising projects aimed at reducing household solid waste (HSW), which accounts for a large portion of MSW. An online questionnaire survey was administered to local governments (LGs) in Japan to research the implementation status of the awareness-raising projects and estimate the waste reduction effect of intervention methods within those projects. Regarding social factors, multiple linear regression analysis showed significant negative relationships of HSW generation rate with the household population, total population, and waste charge system. Conversely, positive relationships were identified with age, the number of cars, income, and the frequency of collection. Intervention methods, such as briefing sessions, utilization of resident leaders, and mobile phone apps, were revealed to be effective; in contrast, information dissemination using the traditional intervention method was not. In particular, the utilization of resident leaders may be the most cost-effective, but some LGs seem to have abolished this form of intervention after its introduction 30 years ago due to lack of empirical evidence supporting its effectiveness.

**Keywords:** household solid waste; questionnaire survey; multilinear regression analysis; intervention; sociodemographic data

## 1. Introduction

### 1.1. Problem of Household Solid Waste and Previous Studies

People's lives are expected to become more affluent because of the worldwide progress of urbanization. As for the waste problems, a positive correlation between urbanization and the per capita municipal solid waste (MSW) generation has been reported in the literature (e.g., [1,2]). Therefore, due to the urbanization of many countries, the total amount of generated MSW is expected to increase far beyond the pace of future population growth. The World Bank [3] has estimated a 19% increase in per capita MSW generation in high-income countries and a 40% increase in low- and middle-income countries by 2050 compared with 2016. This has sounded the alarm that MSW generation in the world could increase by 70% from 2016. Currently, administrative agencies collect and adequately process almost all MSW in high-income countries, whereas the collection rate in low-income countries is as low as 39%, and uncollected waste is openly dumped [3]. Therefore, future increases in MSW generation will lead to an increase in the administrative cost for MSW collection and processing in high-income countries, along with negative impacts on the environment and health in low-income countries. One of the common global issues is now the reduction of MSW generation.

The MSW generation per capita in high-income countries is generally higher than in lower-income countries. While being a high-income country, Japan has a significantly

lower MSW generation of 0.95 kg/capita/day than average for high-income countries, which is 1.57 kg/capita/day [3]. Yet, there was a period when the value increased to 1.19 kg/capita/day, peaking in 2000 during the growth of the economy. Since then, the MSW generation rate has been on a downward trend [4]. This is due to economic recession, but would also not have been possible without the policy efforts of the basic local governments (LGs) responsible for the processing of MSW. As an easy-to-understand indicator, they progressively introduced waste charge systems for household solid waste (HSW), which accounts for a large portion of MSW, ranging from 19.5% for all basic LGs in 2000 to 58.8% in 2021 [5]. Additionally, they also carried out various awareness-raising projects to call on residents to reduce HSW [6,7]. However, a large difference in the generation rates exist among basic LGs, with a maximum disparity of 553 g/person/day (a minimum of 384 and a maximum of 937 g/person/day) [8]. The processing cost corresponding to this disparity has been calculated at about 14.1 million USD/year under the condition that the average unit price for this process in 2019 was 0.349 USD/kg and the average population of LGs was 200,000 residents [8]. The average annual total budget of general LGs was 585.7 million USD [9]; this cost, therefore, cannot be ignored, especially for LGs with substantial waste. Considering that the cost of the same waste process can also be as high as 4–19% of the total budget of LGs around the world [3], the search for good policy practices to reduce HSW generation has become a global issue and challenge.

To reduce HSW, LGs in high-income countries have not only introduced waste charge systems but also carried out various public awareness-raising projects. Compared with the many studies on the former, research on the latter is currently insufficient. However, in recent years, knowledge, especially of awareness-raising projects to reduce food waste, has gradually begun to be compiled around the world. The Waste and Resources Action Program (WRAP) in the UK [10] reported that interventions among citizens to discuss actions against food loss brought about a 50% reduction in food loss. In Canada, sending five e-mails within two weeks led to a reduction in food loss at the household level through general food loss countermeasures, such as shopping and cooking, together with information to inform people about the economic loss of 600 USD/household/year due to food waste [11]. In Japan, Yamakawa et al. [12] report that the distribution of awareness-raising pamphlets by LGs reduced the HSW generation rate, and dietary education and environmental education in elementary schools led to a reduction in food waste. In addition, waste reduction promotion projects, in which some residents are commissioned as leaders by the LGs to disseminate various waste-reducing tips to other residents, were suggested to be effective [13].

At this time, we can see much more variety in awareness-raising projects beyond the above-mentioned projects on the official websites of LGs in many countries. However studies on waste reduction effect by those projects have just begun and knowledge is still lacking, as also pointed out by Stockli et al. [14]. For an advanced waste management policy, the waste reduction effect of each of the various awareness-raising projects need to be unveiled. In particular, focusing not on single project but on various and multiple projects simultaneously should become more crucial because LGs generally carry out a number of projects rather than just one. Of course, there is currently no study focusing on the effect of various projects simultaneously.

### 1.2. Aim of This Study

This study was aimed at estimating the effects of various awareness-raising projects by comparative analysis. Firstly, in order to determine the implementation status of those projects in LGs in Japan, we administered an online questionnaire survey to them. To obtain comparable data to analyze the difference among LGs, we used choices for the types of intervention methods applied in the awareness-raising projects, the types of which types were generalized in accordance with our previous study [6,7]. Secondly, to estimate the waste reduction effect of each intervention method, we analyzed the relationship between the implementation status of intervention methods and per capita HSW generation at LGs

using multiple linear regression (MLR) analysis. In this analysis, we controlled for the effects of social factors such as household population and income.

The scope of waste covered by this study involved all types of HSW, which included not only food waste but also other burnable, recyclable, unburnable and oversized waste, which also place a burden on the environment. Thus, we emphasized the overall reduction in HSW generation.

## 2. Materials and Methods

### 2.1. Questionnaire Survey

If effective intervention methods exist, a significant relationship between their implementation status and per capita HSW generation should be observed. In this case, effective intervention methods should be more common in LGs with low per capita HSW generation ($LG_{low}$) than LGs with high per capita HSW generation ($LG_{high}$). In order to determine the intervention method characteristics for $LG_{low}$, the questionnaire items were designed as shown in Table 1 and we requested a person in charge of HSW reduction for each LG to respond to it. Panel data of awareness-raising projects over multiple years are ideal for evaluating the effects, but would become messy for respondents to answer because they would need to check many previous administrative documents. Accordingly, the questionnaire collection rate or the reliability of answers might have been reduced. To facilitate the collection of answers, we simply asked about the implementation of interventions during the last five years (2017–2021).

**Table 1.** Questionnaire items related to administrative projects for HSW reduction.

| Questionnaire Items | Answer Type |
|---|---|
| Q.1: Has your government office implemented any project from 2017 to 2021? | Selective answer: yes/no (If "yes", please answer the following questions) |
| Q.2: How many people per year were in charge of all the projects in the last 5 years? | Description of number |
| Q.3: Please select all applicable intervention methods for all projects in the last 5 years. (Multiple selections are possible) | Selective answer (see Table A1 for choices M1-i–M8-ii) |
| Q.4: For each specific project in the last 5 years, please provide some information. In the case of multiple projects, please choose up to five projects with the greatest waste reduction effectiveness. | |
| (A) Please enter the first project's name, which you consider the most effective for waste reduction. | (A) Free description |
| (B) Please select all waste types targeted for reduction by the specific project. (Multiple selections are possible.) | (B) Selective answer (see choices in Table A2) |
| (C) (Please select all applicable intervention methods for the specific project. (Multiple selections are possible.) | (C) Selective answer (see choices in Table A1) |
| (D) Please provide any additional information about the project. | (D) Free description |
| (E) Please answer the approximate manpower (how many people over how many days) needed per year to manage the project. | (E) Description of number |
| (The above items are repeated for a maximum of five projects) | |
| (A) Please enter the second project's name, which you consider the second most effective for waste reduction. | (A) Free description |
| . . . | . . . |
| Q.5: How many projects have you managed in the last 5 years? (In cases where five specific projects were provided.) | Selective answer (6–10, 11–15, 16–20, 21–25, 26–30, 31–40, 41–50, and 51 or more) |

First, we asked whether awareness-raising projects had been implemented, and if so, what were the types of intervention methods adopted in all awareness-raising projects (hereafter referred to as all projects). Table A1 is the selective items, reflecting the real

types of intervention methods of LGs in Japan, as derived from our previous study [6,7]. Seven of the major types (M1, M2, M3, M4, M6, M7, and M8 in Table A1) were reported by Stockli et al. [14], while M5 was reported by the Japanese Ministry of the Environment (JMOE) [15]. Specifically, M1 (one-way information flow) and M2 (communication) were integrated as simply "informational" in their report [14]; however, we considered these separately. The LGs also answered questions about up to five specific awareness-raising projects (hereafter referred to as specific projects) they considered effective. They also indicated the waste type targeted for reduction by these projects (see Table A2 for the waste type statistics in Japan [8]). Lastly, the respondents were asked to describe the name of the project and provide additional explanation, such as the annual manpower needed to manage the project. The percentage of manpower for each intervention method was calculated by the following process. First, the annual manpower (people × days) derived from (E) of Q.4 in Table 1 was divided by the manpower of all persons in charge of LGs derived from Q.2 in Table 1 (annual working days = 245 days).

For pretest to confirm whether the question items were easy to answer, we firstly developed question items on Word format (Microsoft) and asked three LGs to respond to it. As a result of the pretest, all respondents were able to claim that the questions were not troublesome to answer; however, they nevertheless missed some questions. To avoid this problem, we decided to use an online questionnaire form—the LoGo Form [16]—which we configured in such a way that respondents would be unable to advance if they did not answer a question. Targeting 540 LGs in Japan with a population of 50,000 or more, the total residents of which cover 84% of the overall population in Japan, we mailed a request letter providing the URL of the LoGo Form to the waste reduction department of each LG on 1 December 2021. We received answers from 405 LGs by the deadline of 1 February 2022 (75% answer rate).

### 2.2. Sociodemographic Data and Waste Collection Policy

The endpoint of the effect of awareness-raising projects probably occurred in the latter half of the project implementation period (2017–2021). The latest data for HSW generation from the period FY2020 were published in April 2022, clearly highlighting the impact of the declaration of a state of emergency related to COVID-19. Thus, we regarded FY2020 as an anomaly and adopted the cross-section data of FY2019, as, due to the COVID-19 pandemic, there was an unprecedented increase in HSW generation linked to the domestic consumption of food at home.

The sociodemographic data and waste collection policy were reported as factors that influence HSW generation. Regarding the former, Gellynck and Verhelst [17] reported a positive correlation of HSW with population density and income, as further supported in other studies ([18,19] and [20], respectively). Tsuzuki et al. [21] reported a negative correlation between HSW and household members. Werf et al. [11] have also reported a significant relationship between food waste and age, household members, and household income.

Regarding waste collection policy, many studies have shown the reduction effect of waste charge systems (e.g., [21–24]). Gellynck et al. [25] have reported that annual waste generation saw a reduction when the weekly collection frequency was cut down. Kurishima [26] has reported that a switch to door-to-door collection reduced waste in Japan, where centralized drop-off points were common. However, the switch was implemented at the same time as the charge system [26].

Referring to the above studies, the control variables for MLR analysis in this study were set as shown in Table 2, where they were obtained from related public data [8,27,28]. We added the workers' ratio in tertiary industries as a proxy variable for urbanization, the total population as a proxy variable for municipality size, and the number of cars per household, which was assumed to affect daily purchasing activities. In total, 48% of the target LGs in this study had introduced waste charge systems. This study did not distinguish the types of charge systems; however, a previous report has revealed that 95% of LGs adopted a unit-based pricing (UBP) system [29]. To avoid multicollinearity between the variables,

correlation coefficients were analyzed in advance, and only representative variables were adopted. For example, the ratios of the population under the age of 15 and over the age of 65 were indicated as factors that influenced HSW generation in a previous study [11]; these variables were correlated with each other and had a high risk of multicollinearity. In this case, only the average household age was adopted as a representative variable. In response to FY2019 of the objective variable for per capita HSW generation, we selected the data of control variables; FY2019–2021. The FY2015 dataset was utilized for the workers' ratio in tertiary industries as it was the most recent.

**Table 2.** HSW statistics and sociodemographic data.

| Variables of Local Governments | Type | Symbol | Statistics (Mean $\pm$ $\sigma$) | Source of Information |
|---|---|---|---|---|
| HSW generation unit (g/capita/day) * | Numeric | $Y_{gen}$ | $651.7 \pm 73$ | National Survey on the State of Discharge and Treatment of Municipal Solid Waste in FY2019 (Japanese Ministry of Environment) [8] * Waste Management Annual Report of Tokyo in FY2019 (Tokyo Metropolitan Government) [27] |
| Frequency of HSW collection (times/week) ** | Numeric | $X_{fc}$ | $2.49 \pm 0.46$ | |
| Waste charge system *** | Categorical (dummy) | $X_{char}$ | $0.48 \pm 0.50$ | |
| Population | Numeric (log-transformed) | $ln(X_{pop})$ | $11.8 \pm 0.8$ | |
| Door-to-door waste collection system | Categorical (dummy) | $X_{dd}$ | $0.09 \pm 0.28$ | |
| Population density (population/km$^2$) | Numeric (log-transformed) | $ln(X_{pd})$ | $6.94 \pm 1.4$ | Population census in FY2020 (Ministry of Internal Affairs and Communications of Japan) [30] **** Population census in FY2015 (Ministry of Internal Affairs and Communications of Japan) [31] |
| Average household age | Numeric | $X_{ha}$ | $47.9 \pm 2.6$ | |
| Household population | Numeric | $X_{hp}$ | $2.32 \pm 0.20$ | |
| Workers ratio in tertiary industries (%) **** | Numeric | $X_{tind}$ | $69.6 \pm 8.7$ | |
| Taxable income per taxpayer (JPY/year) | Numeric (log-transformed) | $ln(X_{inc})$ | $8.07 \pm 0.17$ | Base tax in FY2019 (Japanese Ministry of Internal Affairs and Communications) [32] |
| Number of cars per household | Numeric | $X_{car}$ | $0.78 \pm 0.21$ | Registered vehicles as of March 2021 (Automobile Inspection and Registration Information Association of Japan) [28] |

* Additional source for 23 special wards in Tokyo. ** Regarding burnable, unburnable, and nonseparated waste. *** The system for only partial waste was set to be equal to the free charge. **** Population census in FY2015 [31]. Note: standard deviation, ($\sigma$).

### 2.3. Multiple Linear Regression and Other Statistical Analysis

To avoid the overfitting effect of too many explanatory variables [33], we limited the number of explanatory variables to 10% of the number of valid answers in our questionnaire survey (404), as described in Section 3.1.1. There were 10 sociodemographic variables and 31 variables related to intervention methods, yielding a total of 41 variables. For the analysis of many explanatory variables, a stepwise method is often introduced to select only significant explanatory variables so as to obtain a robust prediction model. However, we did not aim to obtain a highly accurate prediction model but to determine what type of intervention methods were correlated with the HSW generation rate. Therefore, we decided to apply the forced entry method, not a stepwise method, for all 39 explanatory variables, except for M8-i and M8-ii, which were thought to be unreliable due to the low number of answers.

Among the control numeric variables, the total population, population density, and income data deviated significantly from the normal distribution with a long right tail (see the dataset in Table S1). Thus, those variables were natural logarithmic transformed and introduced to MLR analysis. The variance inflation factor (VIF) was used as an index to

confirm the multicollinearity. A wide range of cutoff values for the VIF can be found in the literature. This study adopted a value of four, which is often used as a theoretical basis such that standard errors are doubled at this point [34].

In this study, the normality test was conducted by Kolmogorov–Smirnov method, and a post hoc Bonferroni multiple-comparison test was undertaken in the case of significance detected by ANOVA. Significance level for the statistic tests was set at 5 % ($\alpha = 0.05$). MLR and other statistical analyses were performed using IBM SPSS Statistics 25 (IBM Corporation, Armonk, NY, USA).

## 3. Results and Discussion

### 3.1. Aggregate of Questionnaire Results

#### 3.1.1. Characteristics of All Projects and Specific Projects

A total of 405 LGs responded to our questionnaire, among which 395 were implementing awareness-raising projects (implementation rate of 98%). However, one of the 395 answers was judged to be invalid because its answer to (B) of Q.4 in Table 1 was "others" but its supplemental descriptions were "no answer" only. Thus, the remaining 404 LGs' answers were taken as valid data. The valid data included data on 975 specific projects in total. Descriptive statistics of the results are shown in Table 3a, and the percentage of LGs implementing each intervention method is shown in Table 3b. To aggregate the data in Table 3b, a dummy variable of 1 was provided when LGs selected a certain intervention choice, and a dummy variable of 0 was provided for no choice. A strong correlation was confirmed between the implementation rate of all projects and specific projects ($r = 0.950$, $p < 0.001$) according to the answers to Q.3 and (C) of Q.4 in Table 1. To review the relationships between the per capita HSW generation, the 404 LGs were divided into three groups ($LG_{low}$, $LG_{middle}$, and $LG_{high}$) according to the 25th and 75th quartile values of the per capita HSW generation (Table 3b).

Q.4 in Table 1 provided information on the effective specific projects for each LG; hence, these projects and their intervention methods should be the focus in the future. Accordingly, both $LG_{low}$ and $LG_{high}$ proposed intervention methods they considered to be effective; however, a truly effective intervention method should be skewed toward $LG_{low}$ after controlling for social factors. Q.3 in Table 1 provided information on all projects; thus, the frequent choice of the same intervention methods should become more apparent in the future, minimizing the bias between $LG_{low}$ and $LG_{high}$. These data should be suitable for understanding the exhaustive status of interventions implemented by LGs in Japan.

According to the characteristics of specific projects and all projects, the effects of intervention methods were determined (in Section 3.1.3), and the results were validated through a comparison with our previous study (in Section 3.1.2), which investigated the official websites of LGs in Japan.

#### 3.1.2. Interventions of All Projects

Figure 1a summarizes the percentages of the major types of intervention methods implemented in all projects. For example, if M1-i, M1-ii, M1-iii, M2-ii, and M2-iii in Table A1 were selected, M1 was compiled as three and M2 as two. M1 (one-way information flow) and M2 (communication) accounted for more than half of the total. When M4 (incentive) and M5 (facilitation of reduction and reuse) were added, the total was over 90%. Thus, these four interventions were the main interventions implemented by LGs in Japan. M1 was the most common, consistent with Stockli et al. [14], who pointed out that the number of information-disseminating interventions was relatively large. The overall characteristic was the same as the results of our previous study (Figure 1b), and there was no significant difference at significance level of 5% in the proportions of the eight categories according to the chi-square test ($p = 0.67$). Therefore, the validities of both results were confirmed.

**Table 3.** (a) Aggregate results of projects implemented by LGs (*n* = 394). (b) Implementation rate of interventions described by LGs in the questionnaire (*n* = 404).

| (a) | | | |
|---|---|---|---|
| | **Total** | **Mean** | **95% Confidence Interval** |
| People in charge *[1] | 2343 | 6.2 | 4.9–7.5 |
| Specific projects | 975 | 2.5 | 2.3–2.6 |
| All projects *[2] | 1246 | 3.2 | 2.8–3.4 |

| (b) | | | | | | | | |
|---|---|---|---|---|---|---|---|---|
| | **All Projects** | | | | **Specific Projects** | | | |
| **Interventions** | **All** | **LG$_{low}$** **(*n* = 101)** | **LG$_{middle}$** **(*n* = 202)** | **LG$_{high}$** **(*n* = 101)** | **All** | **LG$_{low}$** **(*n* = 101)** | **LG$_{middle}$** **(*n* = 202)** | **LG$_{high}$** **(*n* = 101)** | **Manpower Rate** **(*M*; Estimate)** |
| M1-i | 94% | 90% | 95% | 95% | 74% | 70% | 75% | 75% | 2.4% |
| M1-ii | 91% | 93% | 91% | 89% | 71% | 68% | 74% | 66% | 2.4% |
| M1-iii | 47% | 49% | 51% | 37% | 29% | 25% | 34% | 24% | 2.9% |
| M1-iv | 26% | 30% | 27% | 22% | 13% | 11% | 15% | 12% | 3.9% |
| M1-v | 66% | 71% | 67% | 57% | 47% | 47% | 50% | 41% | 2.9% |
| M1-vi | 37% | 38% | 40% | 31% | 22% | 20% | 24% | 21% | 3.4% |
| M2-i | 65% | 68% | 67% | 58% | 43% | 47% | 44% | 37% | 3.3% |
| M2-ii | 48% | 48% | 52% | 38% | 31% | 34% | 33% | 24% | 3.7% |
| M2-iii | 59% | 59% | 63% | 50% | 35% | 34% | 39% | 27% | 3.3% |
| M2-iv | 9.9% | 8.9% | 12% | 6.9% | 5.2% | 4.0% | 5.9% | 5.0% | 2.2% |
| M2-v | 25% | 28% | 26% | 20% | 12% | 11% | 14% | 7.9% | 5.4% |
| M2-vi | 61% | 62% | 64% | 52% | 21% | 22% | 20% | 21% | 3.3% |
| M3-i | 6.4% | 5.0% | 7.4% | 5.9% | 3.2% | 4.0% | 3.5% | 2.0% | 8.1% |
| M3-ii | 27% | 34% | 26% | 25% | 14% | 22% * | 12% | 8.9% | 2.8% |
| M3-iii | 15% | 13% | 17% | 14% | 8.7% | 8.9% | 9.4% | 6.9% | 4.5% |
| M4-i | 31% | 39% | 33% | 22% * | 9.9% | 9.9% | 12% | 5.0% | 2.9% |
| M4-ii | 69% | 66% | 69% | 70% | 46% | 47% | 44% | 50% | 2.1% |
| M4-iii | 34% | 36% | 32% | 35% | 21% | 24% | 20% | 20% | 2.5% |
| M4-iv | 7.7% | 9.9% | 6.9% | 6.9% | 3.5% | 4.0% | 3.0% | 4.0% | 4.3% |
| M5-i | 30% | 30% | 32% | 27% | 9.9% | 9.9% | 9.9% | 9.9% | 2.8% |
| M5-ii | 26% | 23% | 31% | 19% | 9.4% | 9.9% | 11% | 5.0% | 2.9% |
| M5-iii | 22% | 15% | 27% | 21% | 4.2% | 2.0% | 6.4% | 2.0% | 1.6% |
| M5-iv | 57% | 54% | 61% | 50% | 25% | 23% | 25% | 28% | 4.0% |
| M5-v | 43% | 47% | 49% | 29% * | 17% | 19% | 18% | 12% | 3.4% |
| M5-vi | 23% | 23% | 25% | 20% | 6.4% | 6.9% | 6.4% | 5.9% | 2.8% |
| M6-i | 10% | 14% | 9.9% | 7.9% | 6.2% | 6.9% | 6.4% | 5.0% | 4.2% |
| M6-ii | 5.9% | 5.9% | 7.4% | 3.0% | 2.5% | 3.0% | 3.0% | 1.0% | 3.7% |
| M7-i | 13% | 11% | 18% * | 6.9% | 6.9% | 5.0% | 9.9% | 3.0% | 3.6% |
| M7-ii | 14% | 12% | 15% | 12% | 5.9% | 6.9% | 6.9% | 3.0% | 5.4% |
| M8-i | 3.7% | 3.0% | 4.0% | 4.0% | 1.5% | 2.0% | 1.0% | 2.0% | 4.5% |
| M8-ii | 2.7% | 1.0% | 2.5% | 5.0% | 1.0% | 0.0% | 1.0% | 2.0% | 1.9% |
| Mean | 34% | 35% | 36% | 30% | 19% | 19% | 21% | 17% | 3.5% |

*[1] Aggregated from 379 LGs that answered. *[2] Aggregated from the median (8, 13, 18, and 23) of actual choices (6–10, 11–15, 16–20, and 21–25). * $p < 0.05$.

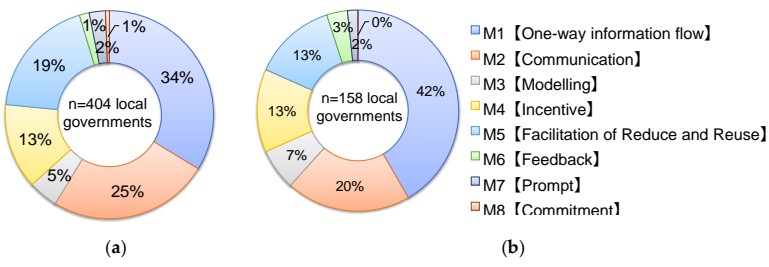

(a)  (b)

M1【One-way information flow】
M2【Communication】
M3【Modelling】
M4【Incentive】
M5【Facilitation of Reduce and Reuse】
M6【Feedback】
M7【Prompt】
M8【Commitment】

**Figure 1.** The implementation percentages of intervention methods for all projects according to major categories: (**a**) questionnaire survey result; (**b**) website survey result [7].

Our previous study [6] suggested that $LG_{low}$ had more diverse intervention types than $LG_{high}$. The average rates of types for all projects in Table 3b were 35% for $LG_{low}$, 36% for $LG_{middle}$, 30% for $LG_{high}$, equivalent to $10.8 \pm 5.0$, $11.3 \pm 5.9$, and $9.4 \pm 5.3$ total types (mean $\pm$ σ), respectively. A significant difference was only observed between $LG_{middle}$ and $LG_{high}$ according to post hoc test with Bonferroni correction ($p = 0.016$) after one-way ANOVA ($p = 0.020$).

As for the details of all projects, M1-i (circular boards, etc.) and M1-ii (official websites, etc.) were particularly high at 90% or more (Table 3b). The former is a traditional means of conveying information in a local community in Japan. The latter is a new method compared with the former, but many LGs currently manage official websites in Japan. While M1-iii (dissemination of information via SNS), M2-v (exchange of views via SNS), and M7-ii (reminder via apps) were also implemented via the internet, the rate was not as high at 47%, 25%, and 14%, respectively. In Japan, LGs have only recently started switching their official websites from PC only to smartphone browsing [35], despite the longstanding popularity of these media with the general public. Similarly, the low implementation percentages of M6–M8 may also reflect their relatively new or unfamiliar nature for LGs. Additionally, as shown in Section 3.1.3, the need for extensive manpower might have also contributed to this low rate.

### 3.1.3. Interventions of Specific Projects

Table 3b shows the manpower percentages for each intervention method. Because the values had many outliers, in order to obtain a representative value we adopted Huber's *M* estimator [36], which is robust against outliers. If only one intervention method was selected for one specific project, the calculated manpower rates could be mapped one to one with the intervention method. However, as multiple intervention methods were selected in many cases, the above-calculated rates represented the sum of manpower for all intervention methods within a specific project. Therefore, the actual manpower might be lower, but the calculated rates were considered to be a guideline for comparing intervention methods. When the correlation between manpower percentages and implementation percentages (Figure 2) was determined, it was not significant at the 5% level ($r = -0.341$, $p = 0.060$), but a tendency toward a negative correlation was observed. According to the analysis, the manpower cost, in addition to the novelty of the intervention method as described above, could influence the difference in the implementation percentages.

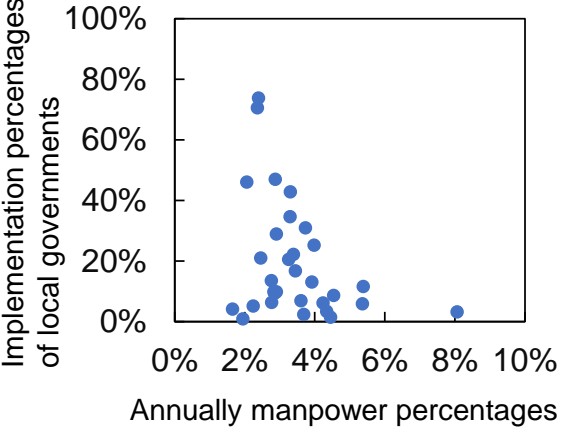

**Figure 2.** Implementation rate of local governments and manpower rate for each intervention.

According to the order of $LG_{high} < LG_{middle} < LG_{low}$ in terms of the implementation percentages in Table 3b, the following intervention methods could be considered effective: M2-i, M2-ii, M3-i, M3-ii, M5-v, and M6-i. Only M3-ii, which had a particularly strong tendency, was significantly different across LG groups in the chi-square test ($p = 0.017$), and the residue analysis of the implementation percentages was significantly higher for $LG_{low}$. As described in Section 3.2.3, the MLR results also suggest the effectiveness of M3-ii.

### 3.1.4. Waste Types Targeted for Reduction

Figure 3a shows the waste types targeted for reduction by the specific projects according to the responses to (B) of Q.4 in Table 1. The ratio excluding all HSW (Table A2) (see Figure 3a) was compared with the composition of HSW discharge (Figure 3b), which was investigated using the JMOE (on a wet weight basis) [37]. According to the results of the JMOE, kitchen garbage and paper were the most common waste types, accounting for more than half of the total. Although the percentages were different, the top two items in Figure 3a,b were coincidental, suggesting that LGs were aware of the actual state of waste discharge. The rate of kitchen garbage was larger than that of paper (Figure 3a), which might have been influenced by the Food Loss Reduction Promotion Act, which was enforced relatively recently in 2019 [38]. The third most common waste type was textiles (11%), such as clothes, but the actual amount of discharge was not high, ranking fifth (3%). In Japan, textiles are still recycled and reused at 22% [39], which is lower than the 45% rate of plastic containers and packaging [40], which ranked third (13%) in Figure 3b, which might have reflected the high awareness of textiles recycle and reuse by LGs. Approximately half of all LGs have collected unused clothes [41]. Furthermore, textile waste has not yet been legislated in Japan [42], in contrast to kitchen garbage, paper, and plastic containers and packaging materials. The political interest in textile waste had been criticized as being insufficient in Japan due to its low amount, harmlessness, minimal scattering, and limited domestic production bases that allow for recycling [39]. However, regulations for the control of textile waste have been strengthened around the world, mainly in Europe [43]. Our survey revealed sufficient awareness of LGs toward textile waste. Therefore, it suggests an opportune time to commence high level discussions on the policy needs for the textile industry in Japan.

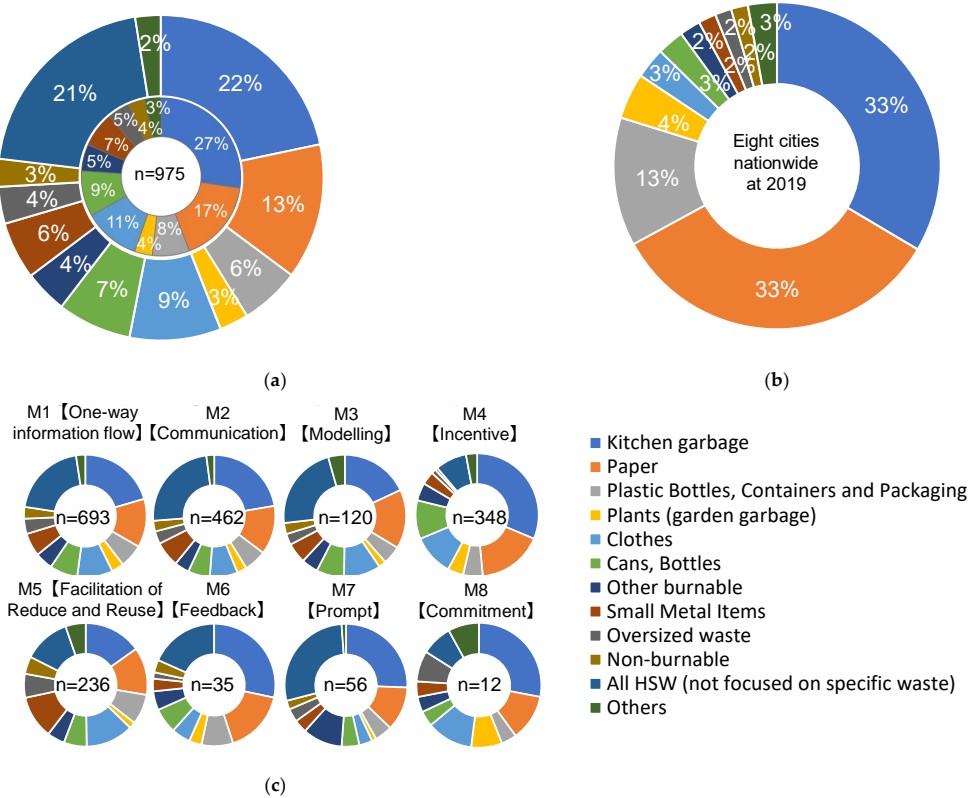

**Figure 3.** Household solid waste types targeted for reduction by the specific awareness-raising projects (**a**) (the proportions inside the figure reflect all types, except for all HSW) and the discharge proportion of waste type on a wet weight basis in Japan (**b**), referred from [37]. (**c**) The proportion of waste types targeted for reduction by major intervention methods; *n* denotes the number of specific projects.

In Figure 3a, all HSW (not focused on specific waste) accounted for 21%, which was approximately the same as the most common type (kitchen garbage, 22%). Figure 3c shows the types of waste targeted for reduction by major intervention methods. The proportion of all HSW was relatively low in M4, M5, and M8. Of these, we did not consider M8 because the representativeness of the data was questionable due to the small number of applicable projects. M4 referred to the utilization of economic rewards and punishments, whereas M5 referred to the preparation of an environment to facilitate reduction and reuse. M4 and M5 were characterized by a system that was designed to stimulate eco-behavior through economic merit or high convenience. The system design of the other six intervention types differed in how they conveyed information to residents, featuring a relatively large proportion of all HSW. In other words, the remaining intervention types did not focus on a specific waste type. Consequently, the intervention methods could be broadly divided based on whether they were targeted at specific waste types. Initially, we predicted that focusing on a specific waste type would help waste reduction; however, the ratio of waste types across $LG_{low}$, $LG_{middle}$, and $LG_{high}$ showed no significant difference according to the chi-square test ($p = 0.855$). Therefore, the data obtained in this study could not confirm this hypothesis.

### 3.2. Multiple Linear Regression (MLR) Analysis

### 3.2.1. Development of the MLR Model

In MLR analysis, we treated the implementation status of each intervention method as an explanatory variable using a dummy variable, along with the sociodemographic data as control variables and the per capita HSW generation as an objective variable (see the dataset in Table S1). The residues of MLR analysis needed to follow a normal distribution, indicating no systematic errors. When the data from all 404 LGs were input, the MLR coefficients were significant ($p < 0.001$); however, the normality of the residues was not observed ($p = 0.041$). The two LGs with the largest residues were numbers 172 and 393 (Figure 4), highlighting the per capita HSW generation. These LGs were the only two to exceed 900 g/capita/day (923.6 g/capita/day and 937.3 g/capita/day, respectively), which were remarkably higher than the 833.7 g/capita/day of the third-ranked, number 343. Comparing all LGs nationwide with a population of 50,000 or more, these two LGs ranked first and third in terms of HSW (the second-ranked LG did not respond to the questionnaire). Upon excluding these two LGs, the residues of the MLR analysis followed a normal distribution (Kolmogorov–Smirnov, $df = 402$, $p = 0.167$), as shown in Table 4, and the multiple regression coefficients were significant ($p < 0.001$). Therefore, for the remaining 402 LGs, the variables could explain the situation of per capita HSW generation. The VIF values were also reasonable since they were <4. Factors other than the variables adopted in this study might be necessary to explain LGs with a remarkably high HSW generation rate. Thus, in the remainder of this paper, the analysis was carried out using the other 402 LGs.

**Table 4.** Multiple linear regression analysis.

|  | Coefficient | β | *p*-Value | VIF |
|---|---|---|---|---|
| Constant | −645.0 | | 0.028 * | |
| $X_{ha}$ | 9.5 | 0.347 | <0.001 *** | 2.2 |
| $X_{hp}$ | −104.8 | −0.305 | <0.001 *** | 3.0 |
| $X_{car}$ | 172.9 | 0.517 | <0.001 *** | 3.3 |
| $ln(X_{pd})$ | −3.6 | −0.073 | 0.382 | 3.9 |
| $ln(X_{inc})$ | 129.5 | 0.317 | <0.001 *** | 2.6 |
| $ln(X_{pop})$ | −15.3 | −0.173 | 0.005 ** | 2.1 |
| $X_{tind}$ | 1.1 | 0.131 | 0.078 | 3.0 |
| $X_{fc}$ | 17.6 | 0.115 | 0.012 * | 1.2 |
| $X_{dd}$ | 8.8 | 0.035 | 0.471 | 1.3 |
| $X_{char}$ | −36.9 | −0.263 | <0.001 *** | 1.3 |

**Table 4.** *Cont.*

| | Coefficient | β | *p*-Value | VIF |
|---|---|---|---|---|
| M1-i | 21.4 | 0.135 | 0.018 * | 1.8 |
| M1-ii | −12.9 | −0.083 | 0.163 | 2.0 |
| M1-iii | 12.3 | 0.079 | 0.129 | 1.5 |
| M1-iv | 5.2 | 0.025 | 0.625 | 1.4 |
| M1-v | 0.2 | 0.002 | 0.976 | 1.5 |
| M1-vi | −6.9 | −0.041 | 0.446 | 1.6 |
| M2-i | −14.5 | −0.102 | 0.042 * | 1.4 |
| M2-ii | 7.9 | 0.052 | 0.316 | 1.5 |
| M2-iii | -0.03 | 0.000 | 0.997 | 1.7 |
| M2-iv | 4.5 | 0.014 | 0.762 | 1.2 |
| M2-v | −7.2 | −0.033 | 0.542 | 1.6 |
| M2-vi | −3.7 | −0.021 | 0.672 | 1.4 |
| M3-i | 10.5 | 0.026 | 0.577 | 1.2 |
| M3-ii | −22.0 | −0.108 | 0.029 * | 1.3 |
| M3-iii | 5.0 | 0.020 | 0.665 | 1.2 |
| M4-i | 3.1 | 0.013 | 0.783 | 1.3 |
| M4-ii | 5.3 | 0.038 | 0.423 | 1.2 |
| M4-iii | 3.1 | 0.018 | 0.696 | 1.2 |
| M4-iv | −3.2 | −0.008 | 0.855 | 1.1 |
| M5-i | 10.7 | 0.046 | 0.375 | 1.4 |
| M5-ii | 4.6 | 0.019 | 0.709 | 1.4 |
| M5-iii | 9.6 | 0.028 | 0.568 | 1.3 |
| M5-iv | 2.0 | 0.012 | 0.795 | 1.2 |
| M5-v | −0.3 | −0.002 | 0.973 | 1.5 |
| M5-vi | 3.2 | 0.011 | 0.816 | 1.3 |
| M6-i | 12.2 | 0.042 | 0.394 | 1.3 |
| M6-ii | 9.9 | 0.022 | 0.644 | 1.2 |
| M7-i | −13.8 | −0.050 | 0.326 | 1.4 |
| M7-ii | −23.0 | −0.078 | 0.100 | 1.2 |
| $_{adj}R^2$ | | 0.274 | | |
| Durbin–Watson | | 1.88 | | |
| *F*-value (*p*-value) | | 4.879 ($p < 0.001$ ***) | | |
| Normality test of residue | | $p = 0.167$ (Kolmogorov–Smirnov) | | |

* $p < 0.050$, ** $p < 0.010$, *** $p < 0.001$.

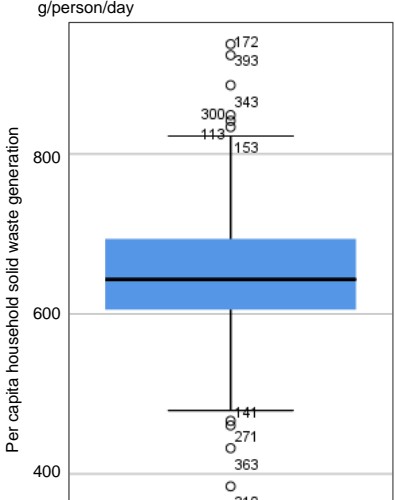

**Figure 4.** Boxplot of per capita household solid waste generation. The upper and lower edges of the box indicate the 75th percentile and 25th percentile, respectively. The upper and lower whiskers indicate 1.5 times the distance of the box height from the median, which is shown as a bold line. Numerical values denote LGs outside of the whiskers.

### 3.2.2. Influence of Sociodemographic Factors

As shown in Table 4, significant negative relationships were observed for the household population ($X_{hp}$), total population ($ln(X_{pop})$), and charge system ($X_{char}$). Conversely, significant positive relationships were observed for the average household age ($X_{ha}$), number of cars per household ($X_{car}$), income ($ln(X_{inc})$), and frequency of collection ($X_{fc}$). As for population density($ln(X_{pd})$, no significant relationships were detected in this study, while the past literature [17–19] has reported a positive correlation with HSW generation rate. As the same, there were no significant relationships with workers' ratio in tertiary industries ($X_{tind}$), and door-to-door collection ($X_{dd}$).

Among the significant variables, we utilized the charge system, which had been introduced by 48% of the target LGs, to validate the obtained MLR model because of the many comparable studies, but the price effect was not considered. The partial regression coefficient of the charge system estimated the waste reduction effect to be −36.9 g/capita/day, equivalent to 4.1% of the 651.7 g/person/day of the HSW generated on average by the LGs. For comparison, we referenced two previous studies regarding the effect on the per capita HSW generation in Japan. Tsuzuki et al. [21] reported the reduction effect of the UBP system at 7.5%. Ichinose [20] estimated the reduction effect to be about 6–8%; however, he did not distinguish the type of charge system. Nevertheless, most LGs have adopted the UBP system.

The waste reduction effect in this study was relatively low compared with previous studies, which was probably caused by the population level of LGs, as previous studies included LGs with fewer than 50,000 residents. The average workers' ratio of primary industries in municipalities with 50,000 or more residents was 5.5%, while in those with fewer than 50,000 residents it was as high as 14.4% in Japan [31]. Yamakawa et al. [44] used the workers' ratio of primary industries as a proxy variable for self-disposal in their analysis and reported a significant relationship with waste reduction. Sekito et al. [45] reported that a higher implementation rate of self-disposal contributed to waste reduction. Therefore, the rate of self-disposal was probably greater in so-called rural areas where primary industries were active. Therefore, the waste reduction effect of the charge system was probably reduced in our study as a function of the population. Accordingly, the obtained MLR model in this study was considered reasonable.

A significant negative correlation has already been described between the household population and per capita HSW generation in Japan [21], as well as in other parts of the world [11,46,47]. However, as discussed below, the relationship between HSW intensity and income or age varies from country to country or region to region. The influence of social backgrounds, such as culture and affluence, might differ according to the region, suggesting that establishing global trends is difficult.

As for age, some reports have shown a negative relationship between food waste and those aged 65 years or older [48–50], whereas other reports have shown opposite results [46,47]. The view of the former studies was that the experience and education of persons during World War II would reduce waste. In contrast, the latter studies considered that the elderly lived in single-person households in many cases, consequently increasing the per capita HSW generation. The current situation in Japan probably fit the latter view. Our study showed a positive correlation between income and HSW intensity, in line with most previous studies (e.g., [17,49,51]), whereas some reports (e.g., [47,52]) have shown the opposite result.

The positive correlation with the frequency of collection in this study does not contradict the report of Gellynck et al. [25], which showed that a lower weekly frequency of collection contributes to reduction of total annual waste generation. As expected, the convenience of taking out garbage increased the HSW.

As for population and cars, we found no previous studies. In this study, the partial regression coefficients showed that a lower population and more cars led to more waste. This characteristic is common in rural Japan, where the per capita ratio of large stores with large parking lots is greater in rural areas than in urban centers [53]. Hence, the

amount of waste might increase as a result of purchasing large quantities using cars at large stores, in line with the study result of Jörissen et al. [46], who revealed that households in Germany and Italy that shop routinely at large supermarkets tend to emit more waste than households that shop in smaller stores or local markets.

### 3.2.3. Influence of Interventions

Significant partial regression coefficients were observed for M1-i (circular boards, etc.), M2-i (briefings, etc.), and M3-ii (waste reduction promoters). The coefficient for M1-i was positive, indicating that the M1-i intervention increased waste, which was not rational. However, this result agreed with Stockli et al. [14], who pointed out that information-disseminating interventions are not effective. They also reported that a combination with other intervention methods could be effective [14]. Thus, in order to investigate the effect of M1-i on boosting M2-i and M3-ii, which showed significant partial regression coefficients, their main effects and interactions were examined using MLR analysis ($F(16, 385) = 11.651$, $p < 0.001$, $adjR^2 = 0.298$, $VIF_{max} = 5.7$). However, there was no significant interactive effect ($\alpha = 0.05$). The effect of other combinations was not examined due to the high risk of multicollinearity, as their VIFs exceeded 4.

M2-i (briefings session, etc.) and M3-ii (waste reduction promoter) promoted a reduction in the HSW. An additional explanation of M2-i in response to (D) of Q.4 in Table 1 was as follows: "Two staff members go to district civic centers and schools and give lectures on how to reduce household waste" (author's translation). This suggested that the administration and residents could exchange opinions via a face-to-face intervention, which was reported to be effective despite the limited number of subjects [10].

The system of M3-ii was added following the revision of the Waste Management and Public Cleansing Law in 1991 in Japan. Residents commissioned by LGs struggled to not only practice waste reduction actions on their own but also to support activities aimed at raising the awareness of other residents. An additional explanation of M3-ii in response to (D) of Q.4 was as follows: "Waste reduction promoters serve as lecturers and hold composting workshops for other citizens in even-numbered months" (author's translation). Another explanation suggested that the projects were aimed at training the promoters. The M3-ii system has a long history, but no research exists outside of the non-profitable organization (NPO) questionnaire survey that was administered to LGs nationwide in 2009 [13], which reported that 34% of all respondents found the system to be effective for waste reduction. Nevertheless, the relationship with HSW generation was not analyzed, and it was unclear on what basis the LGs answered the questionnaire. Their survey revealed that 61% of the 505 LGs had established projects with waste reduction promoters in 2009, whereas, in our 2021 survey, this number dropped to 42% of the 404 LGs (the sum of M3-ii and M3-iii in all projects in Table 3b, corresponding to the NPO survey). Strictly speaking, the two survey methods were not the same, but some LGs might have abolished the projects in the 30 years after the start of the system. However, the cascade training approach (multistage approach) of WRAP in the UK [48], similar to the project in this study, was also reported to be effective at reducing waste. Therefore, this intervention method should be revisited.

M7-ii (language prompt) was not significant but showed a tendency toward a negative relationship with HSW. A total of 27 specific projects applied to M7-ii, of which 10 used mobile phone apps (LINE and Sanaaru [54]). An additional explanation of M7-ii in response to (D) of Q.4 was as follows: "We are disseminating various information such as waste reduction projects and information on various events to apps users" (author's translation). The information distributed by M7-ii was no different from circular boards or websites but may reach more residents because mobile phone apps are easily accessible. In Canada, intervention via e-mails repeatedly sent to citizens achieved a reduction in food loss [11]. Accordingly, M7-ii might be effective for waste reduction. However, a project using such apps is still too advanced for LGs, as the penetration rate of apps in all LGs was only

10% [54]. There were no examples in the research, but it was expected that such apps will spread to other LGs in Japan with the advance of digital transformation.

*3.3. Cost-Benefit Analysis of Interventions*

M2-i (briefing session, etc.), M3-ii (waste reduction promoter), and M7-ii (language prompt) were indicated to be effective for waste reduction; thus, their cost-benefit was calculated using the manpower data in Table 3b. From the respective partial regression coefficients, the reduction effect on the per capita HSW generation was estimated to be −14.5 g/person/day for M2-i, −22.0 g/person/day for M3-ii, and −23.0 g/person/day for M7-ii. The economic effects were calculated under the same conditions as those outlined in Section 1.1, i.e., the average unit price of the general waste process was 0.349 USD/kg, and the average population of LGs was 200,000 [55]. As a result, benefits of approximately 370,000 USD/year, 560,000 USD/year, and 590,000 USD/year were estimated, respectively.

To estimate the manpower cost, the annual manpower requirement was first calculated by multiplying the average staff number of 6.2 (Table 3a) by the manpower rate for each intervention method (Table 3b). Next, the value was multiplied by the average annual salary of LG employees in 2019, i.e., 41,000 USD/person/year. As a result, 8400 USD/year is required for M2-i, 7100 USD/year is required for M3-ii, and 13,800 USD/year is required for M7-ii.

When taking into account the personnel costs, but without the project cost, the three approaches can be considered cost-effective, particularly M3-ii. If the floating manpower obtained from scrapping M1-i, which may not have a waste reduction effect as previously described, is allocated to these three interventions, the administrative management could become more effective and efficient. Naturally, caution is required when concluding that the effects of these intervention are probable. For the future, related knowledge needs to be compiled by more studies and comparisons made with other similar studies using purchasing power parity (PPP) as the basis for a cost comparison of the effects of the interventions shown in this study.

## 4. Conclusions

In this study, we clarified the implementation status of intervention methods aimed at reducing HSW by LGs in Japan. MLR analysis was performed for the estimation of the waste reduction effects. The following findings were obtained:

I. Regarding the implementation rate of the LGs according to the major types of intervention methods, M1 (one-way information flow, 34%) and M2 (communication, 25%) accounted for more than half of the total. When also considering M4 (incentive, 13%) and M5 (facilitation of reduction and reuse, 19%), the total exceeded 90%, indicating that these four were the main intervention methods in Japan.

II. The waste types that the LGs especially targeted for reduction were kitchen garbage, paper, and clothes. On the other hand, there were many specific projects not focused on any waste types. At first, we predicted how strategies that emphasized waste types would affect the result of HSW reduction, but there was no difference in the ratio of waste types across the three LG groups according to the HSW generation rate.

III. Regarding the social factors, we found significant negative relationships between HSW generation rate and household population, total population, and waste charge system, and significant positive relationships with average household age, number of cars per households, income, and frequency of waste collection. Almost of these relationships were consistent with previous studies. The relationship of population and cars indicate that shopping methods led to increased waste generation in rural areas.

IV. Regarding the intervention methods, M1-i (circular boards, etc.) had a significant relationship with heavy waste, M2-i (briefing sessions, etc.) and M3-ii (waste

reduction promoters) had significant relationships with low waste, and M7-ii (language prompt) showed a tendency toward a negative correlation with waste. M2-i was mainly a face-to-face resident-briefing session, whose effectiveness was reported by previous studies despite the limited number of participants. The effectiveness of M3-ii was also indicated even when considering the success of WRAP's cascade training. M7-ii mainly featured the utilization of mobile phone apps, which can directly disseminate information to individuals in contrast to the conventional M1-i intervention.

V.  The operating costs of M2-i, M3-ii, and M7-ii were estimated to be 8400 USD/year, 7100 USD/year, and 13,800 USD/year, respectively, according to the manpower for management, but project costs were not taken into account. The benefits associated with these waste reduction effects are expected to be 370,000 USD/year, 560,000 USD/year, and 590,000 USD/year, with M3-ii considered particularly cost-effective. However, some LGs seemed to have abolished this intervention in the 30 years in which it was started, due to a lack of supporting empirical evidence.

## 5. Limitations of This Study and Suggestions for Future Research

This was a screening study to identify useful intervention methods among a wide variety of applications aimed at a reduction in HSW generation rates. It did not reveal the details of the strategies, such as the implementation period (summer vs. winter, moving in vs. spring cleaning), the number of implementations, the target audience, the audience size, and the informational content to be conveyed. In the future, it will be necessary to conduct interviews and/or questionnaire surveys to investigate these details.

In addition, this study collected cross-sectional data regarding the implementation status of the intervention methods. Therefore, the possibility of reverse causality cannot be ruled out, as many intervention methods were implemented simultaneously to address excessive HSW generation. Effective intervention methods should lead to a reduction in the per capita HSW generation over time. To investigate this effect, panel data that capture changes in the implementation status of intervention methods over time is required. If questions on awareness-raising projects are added to the annual national survey conducted by the JMOE, panel data must be constructed.

The above-proposed future efforts would enable large-scale randomized controlled trials (RCTs), which can be used to accurately analyze the waste reduction effect and promote and legislate evidence-based policies related to awareness-raising projects.

**Supplementary Materials:** The following supporting information can be downloaded at https://www.mdpi.com/article/10.3390/su142214835/s1: Table S1. The dataset for multiple linear regression analysis by Saitoh et al.

**Author Contributions:** Conceptualization, Y.S.; methodology, Y.S., H.T. and A.I.; validation, Y.S. and A.I.; formal analysis, Y.S.; investigation, Y.S.; data curation, Y.S.; writing—original draft preparation, Y.S.; writing—review and editing, Y.S., H.T., K.K. and A.I.; visualization, Y.S.; project administration, Y.S.; funding acquisition, Y.S. All authors have read and agreed to the published version of the manuscript.

**Funding:** This research was funded by JSPS KAKENHI, grant number 20K20033.

**Institutional Review Board Statement:** Institutional Ethics Committee of Gunma Prefectural Institute of Public Health and Environmental Sciences approved Ethical review to be waived for this study due to the lack of personal information and the lack of real intervention against humans (protocol code 30056-12, date of approval 13 October 2022).

**Informed Consent Statement:** Not applicable.

**Data Availability Statement:** In this study, the public statistical data shown in Table 2 were obtained from the Ministry of Japan and various Japanese associations (see sources in the References).

**Acknowledgments:** We appreciate all local government officers who answered this questionnaire survey. We also acknowledge Yuichi Ishimura who provided advice and the contact information of the officers, Sayumi Kanai who supported the questionnaire survey, and Sakiko Someya who supported the visualization improvement of this manuscript.

**Conflicts of Interest:** The authors declare no conflict of interest. The funders had no role in the design of the study; in the collection, analyses, or interpretation of data; in the writing of the manuscript; or in the decision to publish the results.

## Appendix A

**Table A1.** Types of intervention methods.

| M1 | **One-Way Information Flow** Send information only from a local government without an opportunity for the residents to convey their opinions. |
|---|---|
| | i    Periodic publications such as circulation boards and PR magazines |
| | ii   Official website of local government |
| | iii  SNS without communication (only from the local government) |
| | iv   DVD rental/video distribution, etc. |
| | v    Handbooks, supplementary readers, manga, picture-story shows, cards, posters, etc. (distribution/rental/download) |
| | vi   Utilization of media such as newspapers, TV, and radio |
| **M2** | **Communication** Allow the opportunity to exchange information and opinions interactively between local governments and residents. |
| | i    Lecture seminars (briefing sessions, etc.) |
| | ii   Practical seminars (environmental learning with experience, etc.) |
| | iii  Exhibits at environmental trade shows, shopfronts, etc. |
| | iv   Requests for waste reduction actions through door-to-door visits |
| | v    Co-consideration with residents (call for ideas via SNS/questionnaire/guide box/opinion) |
| | vi   Utilization of waste-related facilities (visit/tour/facility rental) |
| **M3** | **Modeling** Disseminate exemplary behavior through influential figures. |
| | i    Celebrities |
| | ii   Citizen leaders (a: HSW-reducing promoter under the Waste Management and Public Cleansing Law) |
| | iii  Citizen leaders (b: other than (a)) |
| **M4** | **Incentive** Promote eco-behavior with opportunity for financial gains and losses. |
| | i    Ideas/poster contests with rewards, etc. |
| | ii   Equipment supply/rental/purchase subsidies (food waste disposer, branch/leaf crusher, etc.) |
| | iii  Reward for eco-behavior (bringing in recyclable waste, etc.) |
| | iv   Economic burden against non-eco-behavior, such as a charge for plastic bags |
| **M5** | **Facilitation of Reduction and Reuse** Prepare equipment and environment to facilitate reduction and reuse. |
| | i    Organize/support flea markets |
| | ii   Government-owned second-hand shops (e.g., Recycle Plaza) |
| | iii  Certification system for second-hand shops, shops for selling by measure, etc. |
| | iv   Equipment preparation at shop and city hall to collect reusable and recyclable waste |
| | v    Foodbank/food drive |
| | vi   Equipment preparation to promote BYO (bring your own) initiatives such as a water dispenser for bottles |
| **M6** | **Feedback** Help to immediately check one's actions and the HSW reduction effect. |
| | i    Food loss diary or similar (issued by the MOE) |
| | ii   HSW diet checklist/HSW loss behavior checklist |

**Table A1.** *Cont.*

| M7 👁 | **Prompt** <br> Call attention to actions working on the five senses (mainly the visual sense). | |
|---|---|---|
| | i | Encouragement to check the refrigerator before shopping using magnets, stickers, etc. (visual prompts) |
| | ii | Reminders by mail, LINE notification, etc. (language prompts) |
| M8 ✋ | **Commitment** <br> Set individual goals for waste reduction. | |
| | i | Choose and swear from an action list presented in advance (selective type) |
| | ii | Swear personal goals without restriction (freestyle) |

**Table A2.** HSW types.

| Waste Type |
|---|
| Kitchen garbage |
| Paper |
| Clothes |
| Plants (garden garbage) |
| Other burnable garbage |
| Plastic bottles, containers, and packaging |
| Cans and bottles |
| Small metal items |
| Oversized waste |
| Nonburnable waste |
| All HSW (not focused on specific waste) |
| Others |

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
