# Peer review of "A Closer Look at Effective Intervention Methods to Reduce Household Solid Waste Generation in Japan"

_sustainability, doi:10.3390/su142214835_

Round 1
Reviewer 1 Report
I proposed a title for consideration by the author. It is short but captures the whole scope of the study.
The intervention methods reported are comprehensive enough but the manner of selecting the respondents through an online survey has not been mentioned.
While the intervention methods were labeled as "effective" no direct metrics on effectiveness were mentioned.
The sentence construction is, at times, very long. There is a need for a technical editor to look into the writing style. All throughout the manuscript, the need to be brief but concise has to be strongly pursued.
The purpose of using stepwise multiple linear regression needs to be clearly stated in the Materials and Methods.
There are sweeping statements and generalizations which need to be carefully stated or modified and should be based on empirical evidence.

Author Response
Dear Reviewer 1,
We appreciate your careful check and important suggestion for our manuscript revision.
Thanks to your detailed check and concrete advice, we think the manuscript was improved to be more straightforward and concisely.
Please see the two attachments in the ZIP file for check our response to your comments (PDF file) and the specific revised points in the current manuscript (Word file). Additionally, we replied to each of your general comments at below in this box.
We are very sorry to make you troublesome too for our revised manuscript.
Thank you.
Best regards,
Y. Saitoh
(・ ; Your comment, ◦ ; Our response)
- I proposed a title for consideration by the author. It is short but captures the whole scope of the study.
- In the revised manuscript we changed the title just as you suggested.
- The intervention methods reported are comprehensive enough but the manner of selecting the respondents through an online survey has not been mentioned.
- In the revised manuscript, we added the detailed information on whom the online questionnaire survey was sent to. That is, we showed clearly that we asked person in charge of HSW reduction in each LGs to answer the questions. In addition to this supplementary, we also explained newly about pretest of the questionnaire survey in accordance to your relevant instruction.
- While the intervention methods were labeled as "effective" no direct metrics on effectiveness were mentioned.
- Firstly, in the Introduction section, we rewrote the definition of “effectiveness” to be more straightforward, that is, at the view point of whether to reduce HSW generation. Regarding the above critic, we thought you had pointed out the section “3.1.3. Interventions of Specific Projects” at the same view point. Just as you instructed concretely there, we should have not used "effect" in this section because there was no comparison with the amount of waste generation but mere comparison among three groups of LGs. Therefore, we recognized the relevant sentence was no necessary and removed it in the revised manuscript.
- The sentence construction is, at times, very long. There is a need for a technical editor to look into the writing style. All throughout the manuscript, the need to be brief but concise has to be strongly pursued.
- At first, we would like to excuse for the tendency of too long sentence construction for the first manuscript. Actually, the first manuscript had been already edited by MDPI English editing service. Unfortunately, there had been almost never specific suggestion at the point of the length of sentence and almost never edit aiming for shorter or more concise too. For example, the below sentence in our draft manuscript, where you pointed out the problem of long sentence, had been edited by the editing service to be the different style but not to be shorter or more concise. As the same for the other sentences too, there had almost never been comments and edit on the length of sentence.
Considering the above situation, at second, we would like to inform you of that the relevant sentences were rewritten for being shorter and concise in accordance to your kind and concrete instructions.
(The first sentence at the second paragraph in 1.1 section.)
Draft sentence: The average of MSW generation per capita in high-income countries is 1.57 kg/capita/day, which is higher than other income level countries, but Japan belonging to the high-income is characteristically lower of that at 0.95 kg/capita/day [1].
Edited sentence: The average MSW generation per capita in high-income countries is 1.57 kg/capita/day, which is higher than in lower-income countries; however, while being a high-income country, Japan has a significantly lower MSW generation of 0.95 kg/capita/day [1].
- The purpose of using stepwise multiple linear regression needs to be clearly stated in the Materials and Methods.
- We would like to appreciate the concrete suggestion for revision and apologize to make you misunderstood. Actually, we applied the forced entry method, not a stepwise method as described in the first manuscript. However, we could reflect on the insufficient explanation in the first manuscript thanks to your critic, and rewrote the relevant points to emphasize that analysis method, just as “we decided to apply the forced entry method not a stepwise method for all 39 explanatory variables”.
- There are sweeping statements and generalizations which need to be carefully stated or modified and should be based on empirical evidence.
- As you criticized, some of the phrases in the first manuscript were a little too exaggerated against the fact. Therefore, we revised those phrases to be more concisely and precisely. For example, the last point in the section 3.3 was revised as below.
“Furthermore, this calculation did not take into account the effects of waste reduction on greenhouse gas reduction. When considering this effect, it would allow for more crucial progress regarding waste management around the world. Naturally, it is required caution to conclude that the effect of these intervention are probable. For the future, related knowledge needs to be compiled by more studies and comparisons made with other similar studies using the purchasing power parity (PPP) as the basis for cost comparison of the effects of the interventions shown in this study. ”

Reviewer 2 Report
This study, explored the implementation status of intervention methods aimed at reducing household solid waste by local governments in Japan through a questionnaire survey. Multiple linear regression analyses were performed to estimate the waste reduction effects of the intervention methods. I accept this manuscript to be published in Sustainability Journal. To improve the manuscript,please consider the following comments:
1- In the Introduction part, the gap that the authors want to be filled, should be clearly indicated then in a clearer way introduce the study aims.
2- What does the sigma symbol (σ) mean in Table 2? add it to the Table footnote.
3- Some of statistical test results was observed in the results part but not mentioned in the the methods section such as normality or multiple comparison (post hoc).
4- The conclusion part is too long, try to shorten it.
Author Response
Dear Reviewer 2,
We appreciate your kind support and important suggestion for our manuscript.
Thanks to your detailed check and concrete advice, we think the manuscript was improved to be more straightforward and concisely.
Please check our responses to your general comments at below in this box, and please see the attachment file of revised manuscript.
We are very sorry to make you troublesome again for our revised manuscript.
Thank you.
Best regards,
Y. Saitoh
(・ ; Your comment, ◦ ; Our response)
- 1- In the Introduction part, the gap that the authors want to be filled, should be clearly indicated then in a clearer way introduce the study aims.
-
- As you indicated for the first manuscript, the description of that gap and this study aim were certainly ambiguous. We thought that ambiguity was caused by no enough description about what of household solid waste (HSW) to reduce and how. Therefore, we rewrote the relevant points to be higher readability for understanding that our study had been aiming for HSW generation reduction. And we added to write that the current gap or research problem we should address was lack of knowledge about reduction effect of the awareness-raising projects. To fill the gap, this study disclosed the implementation status of the projects in local governments in Japan and analyzed the relationship between that status and HSW generation. In the revised manuscript, we newly explained the above content.
- 2- What does the sigma symbol (σ) mean in Table 2? add it to the Table footnote.
- Thank you for the concrete instruction. The meaning of σ was written at Table 2 footnote as “Note; Standard deviation (σ)”.
- 3- Some of statistical test results was observed in the results part but not mentioned in the methods section such as normality or multiple comparison (post hoc).
- Thanks to this concrete suggestion, we think appropriate explanation was newly added in the 2.3 section regarding methodology as the following. “In this study, the normality test was conducted by Kolmogorov–Smirnov method, and post hoc Bonferroni multiple-comparison test was done in the case of significancy detected by ANOVA. Significance level for the statistic tests were set 5 % (α= 0.05). MLR and other statistical analysis were performed using IBM SPSS Statistics 25 (IBM Corporation, Armonk, NY, USA).”
- 4- The conclusion part is too long, try to shorten it.
- We would like to appreciate to your important suggestion too, which gave us chance to review our own manuscript again. Absolutely, just as you pointed out, the conclusion part was too long at first, so we made effort to rewrite it to be shorter but straightforward by highlighted only the major findings of this study. As a result of this revision, 26% of the number of total words in the Conclusion section was reduced.
